# *Gma-miR408* Enhances Soybean Cyst Nematode Susceptibility by Suppressing Reactive Oxygen Species Accumulation

**DOI:** 10.3390/ijms232214022

**Published:** 2022-11-14

**Authors:** Yaxing Feng, Nawei Qi, Piao Lei, Yuanyuan Wang, Yuanhu Xuan, Xiaoyu Liu, Haiyan Fan, Lijie Chen, Yuxi Duan, Xiaofeng Zhu

**Affiliations:** 1Nematology Institute of Northern China, Shenyang Agricultural University, Shenyang 110866, China; 2College of Plant Protection, Shenyang Agricultural University, Shenyang 110866, China; 3Institute of Applied Ecology, Chinese Academy of Sciences, Shenyang 110016, China; 4College of Bioscience and Biotechnology, Shenyang Agricultural University, Shenyang 110866, China; 5College of Science, Shenyang Agricultural University, Shenyang 110866, China

**Keywords:** *miR408*, Soybean cyst nematode, Overexpress, Short tandem target mimic, ROS-reactive oxygen species

## Abstract

Soybean cyst nematode (SCN, *Heterodera glycine*) is a serious damaging disease in soybean worldwide, thus resulting in severe yield losses. MicroRNA408 (*miR408*) is an ancient and highly conserved miRNA involved in regulating plant growth, development, biotic and abiotic stress response. Here, we analyzed the evolution of *miR408* in plants and verified four *miR408* members in *Glycine max*. In the current research, highly upregulated *gma-miR408* expressing was detected during nematode migration and syncytium formation response to soybean cyst nematode infection. Overexpressing and silencing *miR408* vectors were transformed to soybean to confirm its potential role in plant and nematode interaction. Significant variations were observed in the MAPK signaling pathway with low OXI1, PR1, and wounding of the overexpressing lines. Overexpressing *miR408* could negatively regulate soybean resistance to SCN by suppressing reactive oxygen species accumulation. Conversely, silencing *miR408* positively regulates soybean resistance to SCN. Overall, *gma-miR408* enhances soybean cyst nematode susceptibility by suppressing reactive oxygen species accumulation.

## 1. Introduction

MicroRNAs are typically endogenous small non-coding RNAs of 20 to 24 nucleotides (nt) in length, which play a crucial role in negatively regulating post-transcriptional gene expression levels [1]. MicroRNA genes encoding miRNAs are transcribed into primary miRNAs (pri-miRNAs) by RNA polymerase II (Pol II), and then these pri-miRNAs are processed into miRNA-5p/miRNA-3p duplexes by the RNase III family enzyme DICERLIKE1 (DCL1). Subsequently, these duplexes are 2′-O-methylated by the methyltransferase HUA ENHANCER1 (HEN1) at the 3′ end. One strand of the duplex is incorporated into ARGONAUTE1 (AGO1) to form an active RNA-induced silencing complex (RISC). MicroRNA/target complementarity guides RISC to target the mRNA slicing. In plants, these small yet impactful miRNAs play an important role in orchestrating various biological processes, such as development, responses to environmental stress, transduction of phytohormone signals, and defenses against plant pathogens [2].

MicroRNA408 (*miR408*) is an ancient and highly conserved miRNA, which is involved in the regulation of plant growth, development and stress response [3]. A total 182 *miR408* precursors (*MIR408*) and 351 mature *miR408* sequences (*miR408*) have been identified from 118 different plants, which are deposited in miRbase [4] and sRNAanno [5]. Among these one hundred and eighteen plants, there are two species of bryophyte, one species of pteridophyte, three species of gymnosperms, and one hundred and twelve species of angiosperms (dicots vs. monocots are 86 vs. 24, respectively). Most *miR408* precursors are produced separately in the two arms. For instance, the *miR408* precursors produce miRNA in the two arms of *Arabidopsis thaliana*, *Oryza sativa*, *Glycine max*, etc. Among them, the members of *miR408*-5p were quite specific, while the members of *miR408*-3p were conservative. *miR408* evolved gradually from bryophytes and played important roles in the evolution of plants to dicotyledonous plants. It may have been the emergence and change of *miR408* that led to the evolution of various higher plant species [3]. *miR408* has been reported to target a variety of blue copper protein members, including those in small blue proteins (phytocyanin family: plantacyanin/PLC, uclacyanin/UCL) and blue oxidases (laccase/LAC). *miR408* regulates the growth and development of different plants by down-regulating its targets, encoding blue copper (Cu) proteins, affects copper homeostasis in the plant cell [6]. In addition, *miR408* improves tolerance to stress by down-regulating target genes and enhancing cellular antioxidants, thereby increasing the antioxidant capacity of plants [7,8,9,10].

Expression of *miR408* gene adapting to the diverse environmental stresses suggests that multiple different transcription factors may be involved in its regulation. Feng et al. [7] reported that *tae-miR408* was positively correlated with the resistance of host plants to abiotic stresses and stripe rust by regulating one of its target genes, the chemocyanin-like protein gene (*TaCLP1*) in wheat. Upon drought stress, expression of *miR408* was up-regulated in *Medicago truncatula* [10] and in *Hordeum vulgare* [11], while reduced *miR408* expression was observed in *Prunus persica* [12] and in *Oryza sativa* [13]. During *Puccinia striiformis* f. sp. *tritici* (Pst) infection, the up-regulation of *miR408* may trigger the lignin biosynthetic pathway as a hypersensitive response (HR) to prevent epidermis rupture. Ma et al. [14] proposed that increasing the expression of *miR408* can reduce reactive oxygen species (ROS) and regulate the target genes encoding Cu-containing proteins, thereby increasing the endogenous availability of Cu for other Cu proteins involved in response to abiotic and biotic stress.

Here, we identified four *miR408*members in *Glycine max*, which are *gma-miR408*a/b/c/d. Detailed expression patterns during plant development were analyzed by expressing *pmiR408*: GUS in soybean roots. Overexpression (OE) and short tandem target mimic (STTM) silence lines were generated to determine the regulatory role of *miR408* on SCN infected response. This study aimed to understand the function of *miR408* in the SCN resistance of soybeans.

## 2. Results

### 2.1. Identification of miR408 in Glycine Max

As a typical multifunctional miRNA, *miR408* family members have been annotated in more than 100 different plants. In order to reveal the distribution and evolution of *miR408* family members in plants, we downloaded all *miR408* precursor and mature sequences of plants from the miRbase database (http://www.mirbase.org/, accessed on 2 July 2021) and sRNAanno database (http://plantsrna.org/, 2 July 2021). Taxonomic and phylogenetic analyses were performed on them, and the results were summarized in Appendix A. Phylogenetic tree was conducted with 182 precursors sequence from 118 species, as shown in Figure 1.

All phylogenetic analysis steps were conducted using PhyloSuite and its plug-in programs [15]. The *miR408* precursor sequences was aligned with MAFFT. Maximum likelihood phylogenies were inferred using IQ-TREE [16] under the TVMe+R5 model for 5000 ultrafast bootstraps. Evolview-v2 was used to visualize the tree, with *Physcomitrella patens* (Bryophyta, Fuariales) as the root of the tree, followed by *Marchantia polymorpha* (Marchantiophyta, Marchantiales), which are also non-vascular plants. A step further is the vascular plant *Selaginella moellendorffii* (Lycopodiophyta, Selaginellales). A total 112 species of Angiosperms and 3 species of Gymnosperms clustered in a clade. Except for a few precursors, most *miR408* in monocotyledonous plants are clustered in one clade, and *miR408* in dicotyledonous plants are clustered in another clade. The dicotyledonous plants are mainly divided into two clades: Asterids and Rosids. This is consistent with plant evolutionary relationships in the APG IV taxonomy of angiosperms.

*miR408* is widely distributed in different plants, among from Bryophyta (represent the first green plants to colonize terrestrial plants) to Angiosperms. It is clear that the *miR408* family is an ancient family of miRNAs; bryophytes may be the evolutionary ancestors of the *miR408* family, which, since then, have been strongly conserved.

*Gma-miR408* family members described in sRNAanno are *gma-miR408*a/b/c/d (Table 1), and the corresponding precursor sequence is pre-*gma-miR408*a/b/c/d (Table 2, GEO Accession: GSM1874239) [17]. The schematic stem-loop structures of pre-gma-miR408a/b/c/d were shown in sRNAanno database. Sequence alignment of the *miR408* precursor revealed 80.71% identity between *A. thaliana* and *G. max*. The mature miRNA of the precursor was in the identical stem arm (Figure 2). This implied that the pre-*gma-miR408* could be processed correctly to form mature *gma-miR408*.

The sequences of *miR408*-5p and *miR408*-3p were aligned, respectively, then displayed by WebLogo [18] (Figure 2). *miR408*-3p is relatively more conserved, and most mature sequences of *miR408*-3p are identical except for the 1st, 5th, 17th, and 24th bases. In contrast, *miR408*-5p was quite specific, with differences in half the bases. The consensus mature *miR408*-3p sequence was 5′-AUGCUACUGCCUCUUCACCUG-3′ and shared high identity from the 2nd to the 24th nucleotide. Mature *gma-miR408*-3p shared the same sequence as the consensus sequence, indicating that *gma-miR408* might play a similar role as in other species. Based on the reported *miR408* in other species (e.g., *ath-miR408*a, *osa-miR408*a) and its higher abundance in plants, *gma-miR408*-3p was used in further study.

### 2.2. Gma-miR408 Response to Heterodera glycines Infection

To determine the potential function of *gma-miR408* in the response to *H. glycines* infection, the expression levels were detected by quantitative real-time PCR (qRT-PCR) among different days post-inoculation (dpi) in the susceptible soybean cultivar Williams 82 (W82), with the non-infected as control (Figure 3). The abundance of *gma-miR408* did not significantly change in the non-infected plants at different time points. Compared with control, the expression level of *miR408* significantly upregulated reaching a maximum of 5.7 times higher. The *miR408* was significantly upregulated in susceptible Williams 82 during the infection process (1 dpi), migration stage (5 dpi) of nematode. Furthermore, the levels of *miR408* revealed no differences between treatment and control at syncytium formation (10 dpi) and syncytium maintenance stage (15 dpi). The dynamic expression of *miR408* during nematode infection showed that *miR408* responded to *H. glycines* infection, indicating that *miR408* may have a role in resistance to SCN migration and syncytium formation.

### 2.3. Expression Pattern of gma-miR408 in Soybean Roots

To explore the expression pattern of the *miR408* family’s response to cyst nematodes infection, the plant binary expression vector *pNINC2GUS* was used to induce W82 to generate transgenic hairy roots. We produced Agrobacterium-mediated transient transformation of hairy roots expressing the *β*-glucuronidase (GUS) reporter gene driven by the *miR408* promoter (*pmiR408*: GUS). *Gmubi* promoter *ubi*:GUS as a positive control. GUS activity of four independent transgenic hairy roots (*pmiR408*a/b/c/d) were assayed under no-infected and *H. glycines* infected conditions (Appendix A). In non-infected plants, GUS staining was only observed in vascular tissues of the root. Under *H. glycines*-infected conditions, GUS activity was observed in the developing syncytium and the whole root (Figure 4). The expression pattern of *miR408* indicated a functional role during the initiation and progression of nematode parasitism.

### 2.4. Induction of gma-miR408 Overexpression and Silencing Transgenic Soybean Hairy Roots

To further investigate the function of *gma-miR408* in the SCN responses, the plant binary expression vector *pNINC2RNA* was used to induce W82 to generate transgenic hairy roots [19,20]. We obtained *miR408*-silencing transgenic plants via Agrobacterium-mediated transformation with miRNA short tandem target mimicry (STTM, these plants were named STTM-*miR408*) and *miR408* overexpression plants with pre-*miR408*a (OE-*miR408*, Appendix A). Confirming the success of our transgenic approaches, we found that the miRNA abundance of *miR408* was decreased in STTM-*miR408* plants (0.3, *p* < 0.05) and increased in OE-*miR408* plants (8.5, *p* < 0.05) compared with their corresponding empty-vector (EV) plants (Figure 5).

### 2.5. Transcriptomic Analyses for OE-miR408 and STTM-miR408

To further explore the molecular basis of resistance against SCN, an RNA sequencing assay was performed using transcriptomic hair roots. The cDNA libraries were generated from 15-day-old transgenic hairy roots induced by *A. rhizogenes* K599. Empty-vector plants were used as control. Three soybean plants were pooled as one sample, which were named EV, OE, and STTM, respectively. The quality of RNA samples was detected using NanoDrop 2000 (Thermo, Waltham, MA, USA). We constructed nine small RNA libraries (3 treatment × 3 biological replicates) sequences using an Illumina novaseq6000 platform. The nine libraries provided a total of 452,070,014 sRNA raw reads, and the Q30 of all the libraries was ≥90.8%. After data quality control, a total of 374,555,456 sRNA clean reads were obtained, and each library contained a minimum of 38,188,248 clean reads. The data were then mapped to the soybean genome. More than 89.6% reads were successfully aligned to the reference genome (Table 3).

### 2.6. Differentially Expressed Genes for OE-miR408 and STTM-miR408

The differentially expressed genes (DEGs) were filtered via DESeq with padj < 0.05 (Table 4, Figure 6). Compared OE with EV, 3653 DEGs were examined including 2191 and 1462 up- and down-regulated genes, respectively (Appendix A). A total of 933 DEGs were selected (598 and 335 DEGs up- and down-regulated between EV and STTM, respectively, Appendix A). Compared OE with STTM, 3831 DEGs were detected including 1626 and 2205 up- and down-regulated genes, respectively (Appendix A). Venn diagrams show 432 DEGs shared by EV vs OE and EV vs STTM (Appendix A). There were 85 and 32 DEGs detected for the opposite regulated between them (Appendix A).

### 2.7. Functional Analysis and Clustering

KEGG pathway and GO ontology analyses were conducted for these DEGs. KEGG pathway enrichment analyses showed that DEGs were mainly mapped into Biosynthesis of secondary metabolites (ko01110), Metabolic pathways (ko01100), MAPK signaling pathway-plant (ko04016), and Plant-pathogen interaction (ko04626), and Starch and sucrose metabolism (ko00500) showed in Figure 7. MAPK signaling pathway-plant (ko04016), and Plant-pathogen interaction (ko04626) were shown in Appendix A.

The results revealed that eight GO terms were enriched commonly and significantly, including response to hypoxia (GO:0001666), response to decreased oxygen levels (GO:0036293), response to oxygen levels (GO:0070482), response to chitin (GO:0010200), cellular response to oxygen levels (GO:0071453), cellular response to decreased oxygen levels (GO:0036294), response to wounding (GO:0009611), cellular response to hypoxia (GO:0071456) showed in Figure 8.

### 2.8. Gma-miR408 Suppressing Reactive Oxygen Species Accumulation

We next investigated ROS levels in the roots of different soybean lines by monitoring the accumulation of ROS (H_2_DCFDA probe labeling) and H_2_O_2_ (3,3′-diaminobenzidine staining, DAB). As expected, the EV plants exhibited an accumulation of both ROS and H_2_O_2_. When compared with EV, OE-*miR408* and STTM-*miR408* transgenic roots had decreased ROS and H_2_O_2_ levels, whereas the ROS levels (fluorescence signal intensity of H_2_DCFDA probe labeling and dark brown stains of DAB staining) were ordered EV > STTM > OE (Figure 9). Thus, we hypothesized that *miR408* expression could inhibit reactive oxygen species accumulation. Meanwhile, *miR408* is not the only factor affecting the accumulation of ROS.

### 2.9. Gma-miR408 Enhances Soybean Cyst Nematode Susceptibility

The regulation of *miR408* in the resistance of soybean to SCN was verified by inoculating soybean cyst nematode on transgenic plants. Three transgenic lines (EV, OE-miR408 and STTM-miR408) were used. Nematode infection and development were detected at 1 dpi and 15 dpi by a modified acid-fuchsin staining-destaining procedure [21].

At 1 dpi, the number of second-stage juveniles (J2s) invading soybean hairy roots reached 149 to 194, and there was no difference in nematode infestation between EV, OE-*miR408* and STTM-*miR408* (Figure 10A). At 5 dpi, obvious differences were observed in the total number of nematodes and swollen juveniles (Figure 10B). For the total number, OE-*miR408* increased significantly compared with 126 nematodes in EV, at 155 (*p* < 0.05). In contrast, the total number of nematodes was significantly reduced, with 86 in STTM-*miR408* (*p* < 0.05). There were 57 swollen juveniles in STTM-*miR408*, and 93 swollen juveniles in EV. STTM-*miR408* had a significant difference with EV (*p* < 0.05). OE-*miR408* had 120 swollen juveniles, which was significantly higher than EV.

At 15 dpi, there were obvious differences in the total number of nematodes and the average number of nematodes in each stage (Figure 10C). The total number of nematodes in OE-*miR408* increased significantly compared to 110 nematodes in EV, at 139 (*p* < 0.05). In contrast, the total number of nematodes was significantly reduced in soybean hairy roots with 70 in STTM-*miR408* (*p* < 0.05). At 15 dpi, the number of J2s in EV, OE-*miR408* and STTM-*miR408* hairy roots was not significantly different. There were significant differences in the numbers of swollen juveniles and females in each hairy root. There were 22 swollen juveniles and 30 female juveniles in STTM-*miR408*, 40 swollen juveniles, and 62 female worms in EV. STTM-*miR408* had significant difference with EV (*p* < 0.05). OE-*miR408* had 48 swollen juveniles and 67 females, which was significantly higher than EV. These results showed that overexpressing *gma-miR408* enhances soybean cyst nematode susceptibility.

## 3. Discussion

MicroRNAs are the second most abundant plant sRNA class [22]. Many plant miRNA families have been conserved for a long time in the evolution of land plants. Axtell and Bartel [23] demonstrated that at least eight miRNA families remained basically unchanged before the emergence of seed plants, and at least two families (*miR160* and *miR390*) have remained unchanged since the last common ancestor of mosses and flowering plants. *miR408* was widely distributed in different plants, from Bryophyta (representing the first green plants to colonize terrestrial plants) [24,25], to ferns that have differentiated vasculature, gymnosperms with uncoated ovules, and angiosperms that flowered and set seed in the end. Evolutionary analysis showed that the *miR408* family was an ancient and widely distributed miRNA family [4,5]. The *miR408* evolution was consistent with plant evolutionary relationships in the APG IV taxonomy of angiosperms.

Since *miR408* was first discovered in *Arabidopsis thaliana*, the crucial roles of *miR408* family members in plants have been verified in many studies [26], and *miR408* has been regarded as an important regulator of plant vegetative growth and reproductive development. In recent years, an increasing number of studies have shown that, in addition to regulating plant growth and development, *miR408* is also stress-responsive in many plant species. Respond to cold stress, the expression of miR408 increased in *A. thaliana* and *O. sativa* [14,27]. *miR408* helps plants improve their tolerance to cold stress by regulating the genes related to Cu homeostasis, oxidative stress, and lignin biosynthesis. Under salinity stress, the up- or down-regulation of *miR408* expression in *A. thaliana*, *Salvia miltiorrhiza*, *Nicotiana benthamiana*, *O. sativa* and *Triticum aestivum* is inconsistent, indicating that different plants have different salt stress resistance mechanisms [7,14,28,29]. *miR408* not only plays important roles in the tolerance to various abiotic stresses, but also plays vital roles in the tolerance to biotic stresses such as LPS stress (Gram-negative bacteria), *Puccinia striiformis* f. sp. *Tritici* (Pst) infection, *P. graminisfsptritici* infection and *Rhizoctonia solani* infection [7,14,30,31,32]. Our study found that nematode infection can induce the expression of *miR408* in roots. During nematode migration and syncytium formation, the expression of *miR408* was significantly up-regulated, indicating *miR408* may have a role in resistance to SCN migration and syncytium formation.

Among the identified roles of *miR408* and its targets in response to biotic and abiotic stress, some have been verified by performing overexpression, T-DNA insertion, and RNAi experimentation. *miR408* is involved in various biotic and abiotic stresses, existing studies have shown that *miR408* has a great relationship with the antioxidant system. Ma et al. [14] proposed that increasing the expression of *miR408* can reduce ROS and regulate the target genes encoding Cu-containing proteins. In *O. Sativa*, a late responsive *miR408* might be involved in *R. solani* infection [31]. Gupta et al. [32] reported that *miR408* in *T. aestivum* was involved in the defense response to stem rust infection. During Pst infection, up-regulation of *miR408* triggers the lignin biosynthetic pathway as a hypersensitive response (HR). Here, we explore the function of *gma-miR408* by overexpressing and silencing *miR408* transgenic plants via Agrobacterium-mediated transformation. Transcriptomic analyses for OE-*miR408* and STTM-*miR408* showed DEGs were mainly involved in MAPK signaling pathway-plant, and Plant-pathogen interaction. The amount of ROS accumulation in both ways was EV > STTM > OE. We hypothesized that *miR408* expression could inhibit reactive oxygen species accumulation. *miR408* maybe not the only factor affecting the accumulation of ROS, exploring the factors and mechanisms that regulate the homeostasis of reactive oxygen species will be the focus of future research.

In conclusion, the results from this study clearly indicate that *gma-miR408* responds to SCN infection, during nematode migration and syncytium formation. Overexpressing *miR408* could negatively regulate soybean resistance to SCN by inhibiting reactive oxygen species accumulation. In reverse, silencing *miR408* positively regulates soybean resistance to SCN. Overall, *gma-miR408* enhances soybean cyst nematode susceptibility by suppressing reactive oxygen species accumulation.

## 4. Materials and Methods

### 4.1. Plant Materials

The cultivated soybean seeds of *Glycine max* (Williams 82, W82) provided by the Nematology Institute of Northeastern China (Shenyang, China) were used for our study. The seeds were surface sterilized with 1% NaClO for at least 10 min and then washed several times with distilled water. The seeds were added to PVC tubes (height × diameter = 10 × 3 cm) containing equal ratios of sterilized sand and soil in a climatic chamber (light/dark = 16/8, 23–26 °C, 50% relative humidity). Hoagland’s nutrient solution was added to soybean seedlings once every three days. Soybean germination and culturing proceeded for 10 days for nematode inoculation and 5–7 days for soybean inoculation with *Agrobacterium rhizogenes* K599 (cotyledons not unfolded).

### 4.2. Nematode

The *Heterodera glycines* (soybean cyst nematode, SCN) race 3 population was propagated on W82 in infected soil in a greenhouse in the Nematode Institute of Northeastern China. SCN cysts were extracted from the infected soil on a 60-mesh sieve (250 µm), and harvested cysts were then crushed on an 80-mesh sieve (180 µm). Eggs were collected on a 500-mesh sieve (25 µm) and further purified using 35% (*w*/*v*) sucrose solution. Eggs were sterilized with 0.1% NaClO for 10 min before being rinsed with sterilized water several times to remove any traces of NaClO. The sterilized eggs were transferred to a modified Baermann pan with 3 mM ZnSO_4_ at 25 °C in the dark for 5 days to allow them to hatch, and the freshly hatched pre-parasitic second-stage juveniles were then harvested.

### 4.3. Nematode Infection

Ten days after soybean seedlings were geminated, 2000 J2s were added to each root system. For hairy soybean roots induced, 500 J2s for each root system. Infections were synchronized by washing the infected roots 24 h post-inoculation.

### 4.4. Construction for Genetic Transformation Vectors

To construct *pmiR408*:GUS, the precursor sequences of soybean *miR408* (*gma-miR408*a/b/c/d) were amplified from the genomic DNA of W82 using the primers a/b/c/dp-F and a/b/c/dp-R, respectively. All the primers used are shown in Appendix A. The PCR products were further added base arm using the primers a/b/c/dp-Fi and a/b/c/dp-Ri, respectively. Then, precursor sequences of soybean miR408 were subcloned into *pNINC2GUS* and verified by primers JCF and JCR (Appendix A). The vector *pNINC2GUS* uses *pCAMBIA3301* as a backbone and combined it with the enhanced CaMV 35S promoter, GUSPlus, and visualization element (EGFP tag) of p4305.1. *Gmubi* promoter *ubi*:GUS as a positive control.

The vector *pNINC2RNAi* uses *pCAMBIA3301* as a backbone and combined with the *Gmubi* promoter, rcbS terminator, and visualization element (EGFP tag) of *pG2RNAi2*. To silence miRNAs, we designed short tandem target mimics (STTM) of *miR408*. STTM-*miR408* consists of two short identical sequences that mimic *miR408* target sites with three additional nucleotides CTA bulges corresponding to positions 10 to 11 of the *miR408*. Two short identical sequences were linked with 48 nt linker and flanked with *Asc* I and *AvR* II restriction enzyme sites at 5′ end and 3′ end, respectively. The promoter of *miR408* and STTM-*miR408* (Appendix A) were synthesized in Genewiz (Wuhan, China); finally, they were cloned into *pNINC2RNAi* and named as *pNINC2EX OE-miR408* and *pNINC2EX STTM-miR408*. Positive validation of vectors transfers into *A. rhizogenes* K599 by primers EGFG F/R and JC F/R (Appendix A).

### 4.5. Transformation Process of Soybean Hairy Roots

GUS, overexpressing, and silencing plasmids were transferred into *A. rhizogenes* K599 by the freeze-thaw method. Then, transgenic hairy roots were induced by the Agrobacterium-mediated method described as [33]. Briefly, young seedlings with unfolded cotyledons are infected with *A. rhizogenes* at the cotyledonary node, and the infection sites are preserved in a humidified environment. After 5–7 days, hairy roots started to sprout from the site of infection. Hairy roots were covered with sterilized vermiculite wetted by Hoagland’s nutrient solution for ten days until hairy roots could sustain the plants. Then, the hairy roots were screened with a handheld lamp (Luyor, Shanghai, China) to visualize GFP expression. Hairy roots carrying strong GFP signals were reserved and used for further tests, and the rest of non-GFP hairy roots and the main roots were removed. After 15 days, transgenic soybean hairy roots were collected to further confirm the effects of overexpressing and silencing.

### 4.6. Gene Expression Analysis of miR408

The samples were collected from roots of 1 dpi, 5 dpi, 10 dpi, and 15dpi in susceptible soybean cultivar Williams 82, with the no-infected as control. For soybean hairy roots, the samples were collected from hairy roots cultured for 15 d, and the effects of over-expression and silencing were confirmed by detecting the expression level of the *miR408* gene by qRT-PCR (the primers used for qRT-PCR are shown in Appendix A). The soybean U6 (GenBank accession LOC100819552) was used as an internal reference gene, with three parallel and three biological repeats per sample, and the results were analyzed by the 2^−ΔΔ*CT*^ method.

### 4.7. RNA Isolation, Reverse Transcription, and cDNA Libraries Sequencing

The total RNA was isolated from the soybean hairy roots using Reagent RNA extraction kit (CWbio, Beijing, China). The total extracted RNA was used to generate sequencing libraries using the NEBNext® Ultra™ RNA Library Prep Kit for Illumina® (NEB, Beijing, China). Library quality was assessed on the Aglient Bioanalyzer 2100 (Agilent, Waldbronn, Germany). The cDNA libraries were sequenced on the Illumina sequencing platform by Metware Biotechnology Co., Ltd. (Wuhan, China).

### 4.8. Differential Gene Analysis and Enrichment

DESeq2 v1.22.1 /edgeR v3.24.3 was used to analyze the differential expression between two groups, and the *p* value was corrected using the Benjamini and Hochberg method. The corrected *p* value and |log2foldchange| are used as the threshold for significant differential expression. The enrichment analysis was performed based on the hypergeometric test. For KEGG, the hypergeometric distribution test is performed with the unit of the pathway; for GO, it is performed based on the GO term.

### 4.9. Statistical Analyses

Statistical analyses were conducted using IBM SPSS STATISTIC v.25 (Armonk, NY, USA) and GraphPad Prism 9 software (GraphPad Inc., San Diego, CA, USA). Multiple t-tests were applied to detect the significant differences in the relative expression level of *miR408* at different times, and one-way analysis of variance (ANOVA) was applied to detect the significant differences in the relative expression level of *miR408* of different hair roots and the number of nematodes.

## 5. Conclusions

In conclusion, the results from this study clearly indicate that *gma-miR408* responds to SCN infection. During nematode migration and syncytium formation, the expression of *miR408* was significantly up-regulated relative to soybean roots compared with control plants. No differences were observed in the later parasitism stage. Nematode infection can induce the expression of *miR408* in whole roots, resulting in GUS signaling, while GUS signal was only expressed within the vasculature of no-infected control. Overexpressing and silencing *miR408* vectors were transformed to soybean to confirm its potential role in plant and nematode interaction. Significant variations were observed in the MAPK signaling pathway with low OXI1, PR1, and wounding of the overexpressing lines. Overexpressing *miR408* could negatively regulate soybean resistance to SCN by suppressing reactive oxygen species accumulation. Conversely, silencing *miR408* positively regulates soybean resistance to SCN. Overall, *gma-miR408* enhances soybean cyst nematode susceptibility by suppressing reactive oxygen species accumulation. 

## Figures and Tables

**Figure 1 ijms-23-14022-f001:**
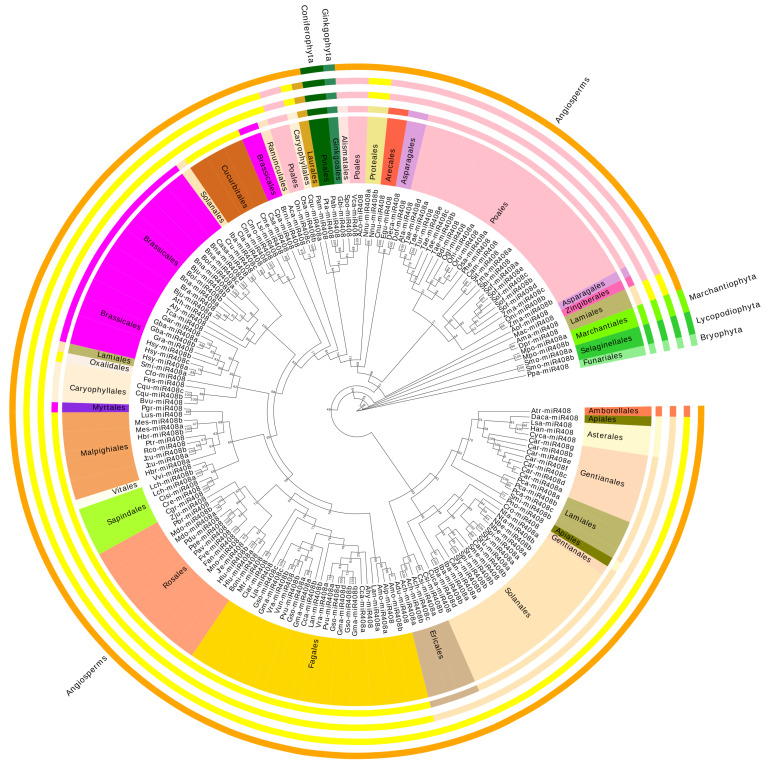
Phylogenetic analysis of *miR408* precursor from 118 plant species. The *miR408* precursor sequences were aligned with MAFFT. Maximum likelihood phylogenies were inferred using IQ-TREE under the TVMe+R5 model for 5000 ultrafast bootstraps. Evolview-v2 visualized the tree. Group labels from outside to inside were division, class, clade, order according to the 4th Angiosperm Phylogeny Group (APG) classification of angiosperms.

**Figure 2 ijms-23-14022-f002:**
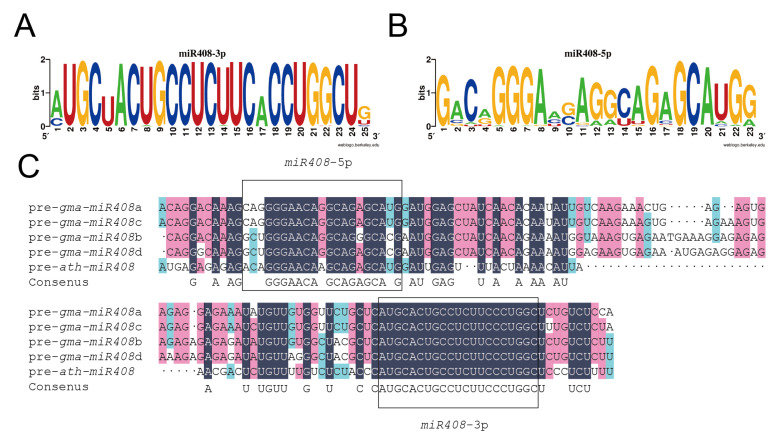
Multiple alignment of *miR408* sequences. (**A**) Consensus sequences of mature *miR408*-3p displayed by WebLogo. (**B**) Consensus sequences of mature *miR408*-5p displayed by WebLogo. (**C**) Precursor sequence analysis of *miR408* between *Glycine max* and *Arabidopsis thaliana*. Sequence alignment of pre-*ath-miR408* and pre-*gma-miR408*a/b/c/d. Boxes indicate the sequences of mature *miR408*-3p and *miR408*-5p.

**Figure 3 ijms-23-14022-f003:**
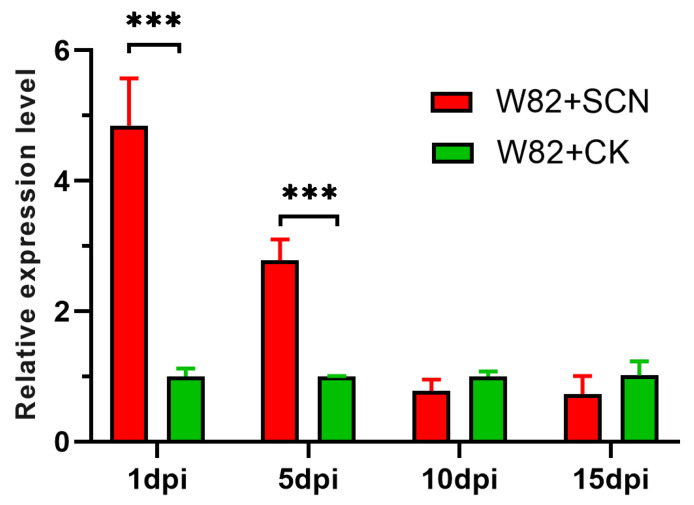
Quantitative Real-time PCR validations of soybean cyst nematode responsive miRNAs in the susceptible soybean cultivar Williams 82. Soybean cyst nematode infected Williams 82 plants were named as W82+SCN, the non-infected as W82+CK. The level of expression was normalized to the level of *U6*. Each bar shows the mean ± SE of triplicate assays. Multiple t-tests were applied to analyze the difference betweeen W82+SCN with W82+CK. *** indicates a statistically significant difference as a relative at *p* < 0.001.

**Figure 4 ijms-23-14022-f004:**
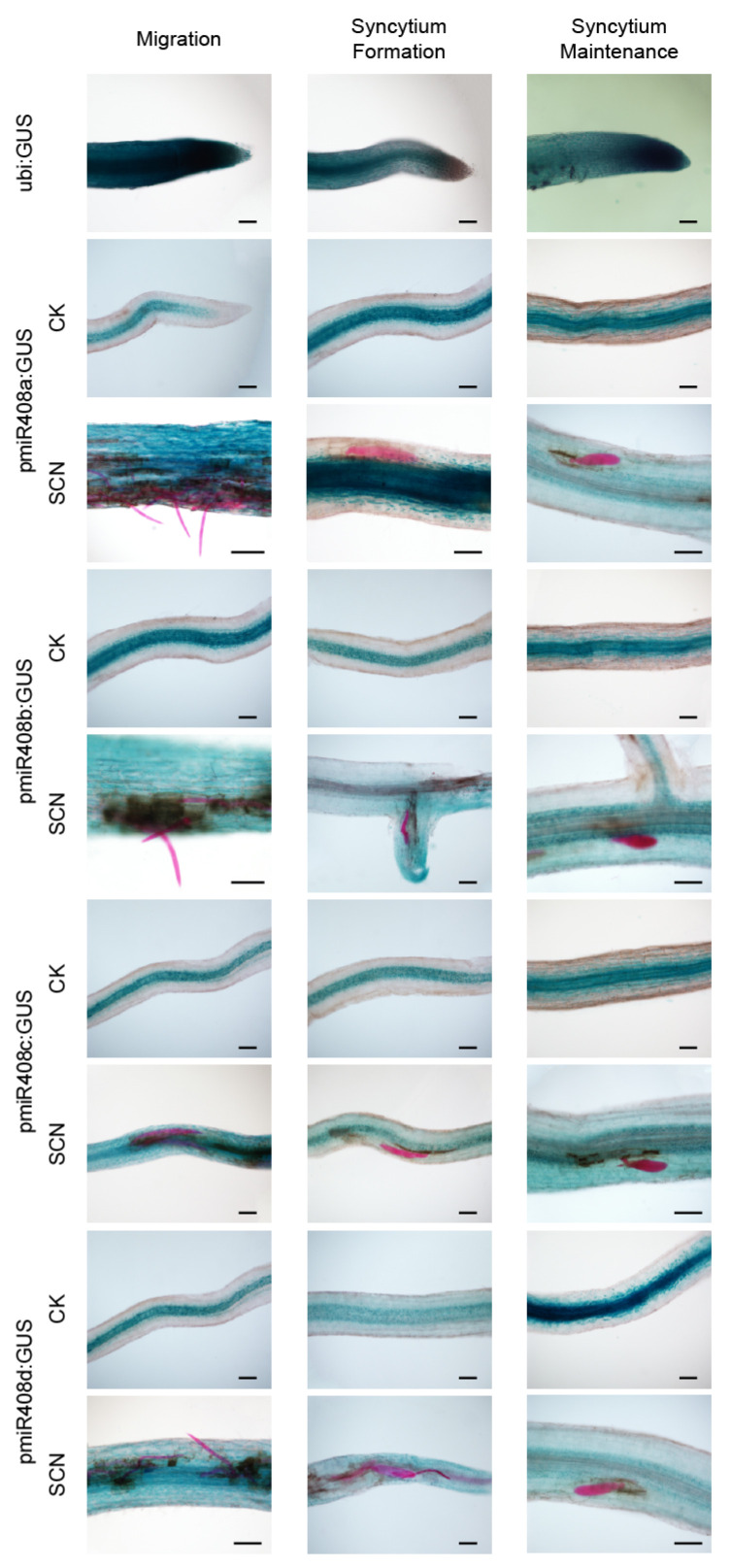
Histochemical staining of GUS activity driven by *miR408* promoters in transgenic soybean in response to *H. glycine* infection. Soybean cyst nematode infected Williams 82 plants were named W82+SCN, the non-infected were W82+CK. Use *ubi*:GUS as a positive control. The *miR408* promoter sequence after adding base arms was inserted into the linearized *pNINC2GUS* and obtained the GUS vector *pmiR408*:GUS. GUS activity of the *pmiR408*:GUS plants under non-infected conditions only showed in the vascular tissue of roots. While the whole root showed GUS activity response to nematode infection. GUS reactivity gradually weakens with the stage of nematode infestation. Bars = 100 µm.

**Figure 5 ijms-23-14022-f005:**
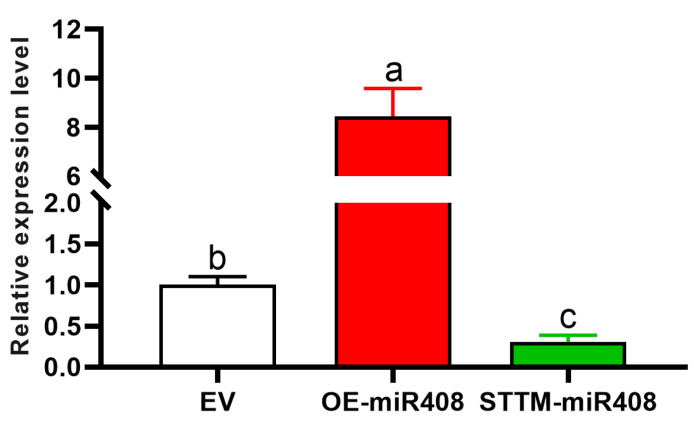
The expression level of *miR408* in transgenic OE-*miR408* and STTM-*miR408* soybean hairy roots. The level of expression was normalized to the level of *U6*. Each bar shows the mean ± SE of triplicate assays. Statistical comparisons between nematodes in EV and other transgenic hairy roots were made by one-way analysis of variance (ANOVA). Different characters mean significant differences found at *p* < 0.05.

**Figure 6 ijms-23-14022-f006:**
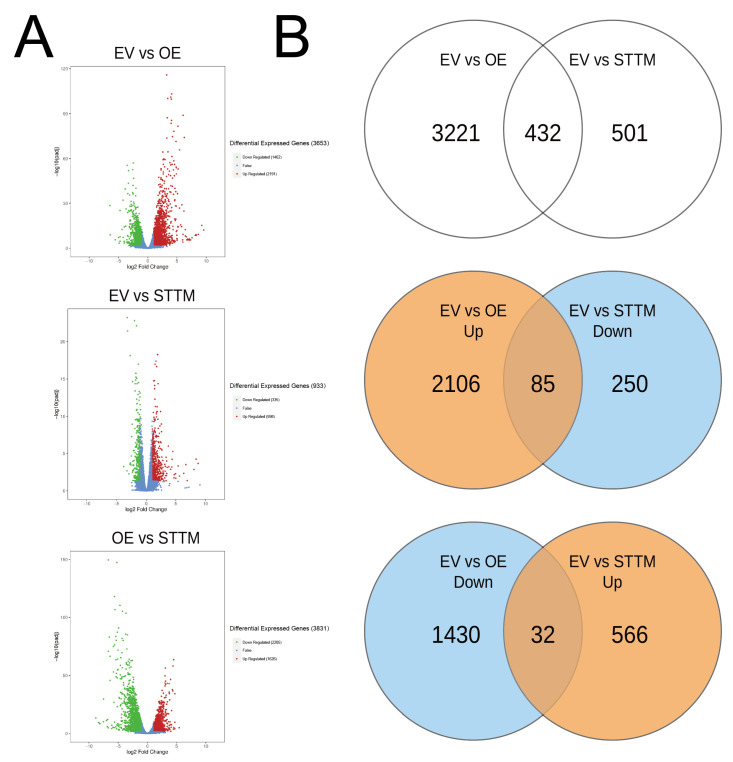
Differential expressed genes analysis in each comparison group. (**A**): The volcano plots show the numbers of significantly differentially expressed genes in each comparison group (padj < 0.05). (**B**): Venn diagrams show the distribution of up and down-regulated genes between EV vs. OE and EV vs. STTM (padj < 0.05).

**Figure 7 ijms-23-14022-f007:**
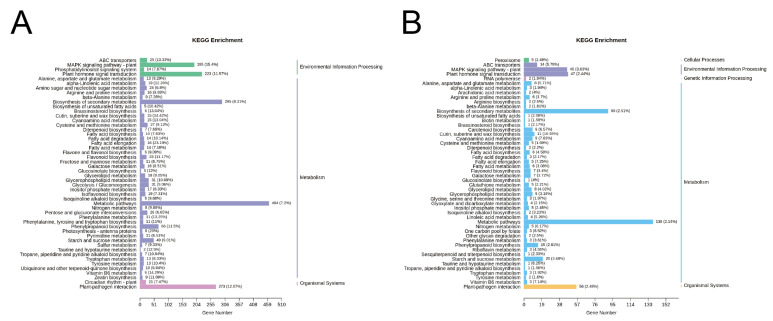
KEGG pathway enrichment of induced DEGs. (**A**) KEGG pathway enrichment of DEGs between EV vs. OE. (**B**) KEGG pathway enrichment of DEGs between EV vs. STTM.

**Figure 8 ijms-23-14022-f008:**
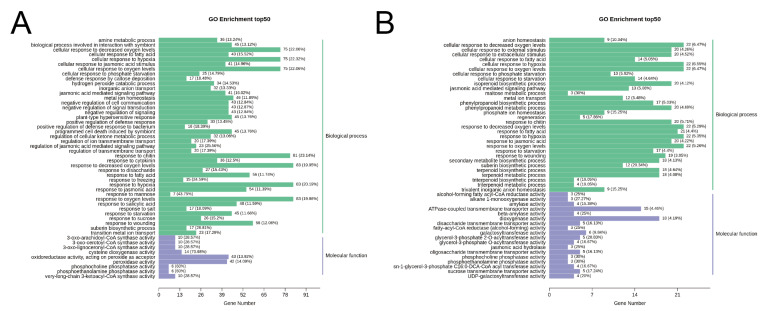
GO enrichment of induced DEGs. (**A**) GO enrichment of DEGs between EV vs. OE. (**B**) GO enrichment of DEGs between EV vs. STTM.

**Figure 9 ijms-23-14022-f009:**
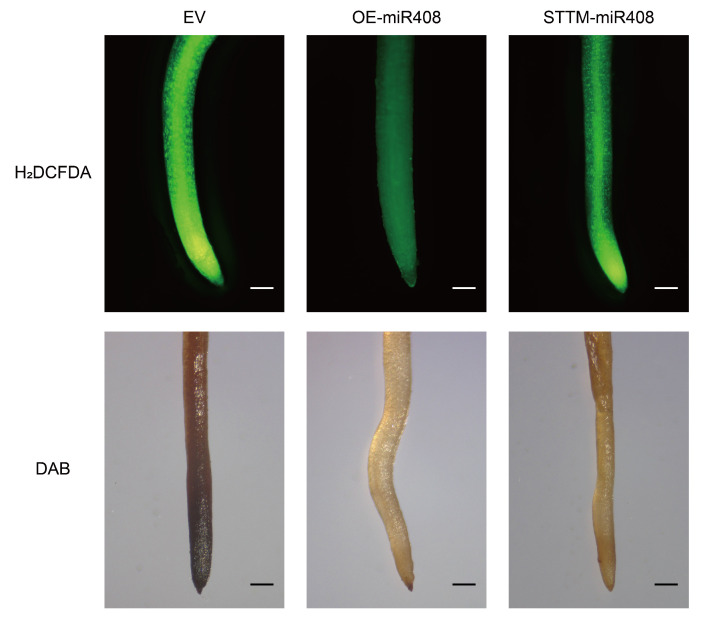
Detection of physiological and biochemical indicators of the OE-*miR408* and STTM-*miR408* soybean hairy roots. H_2_DCFDA probe labeling and DAB staining of Agrobacterium-mediated transient transformation of hairy roots. Bar = 100 µm.

**Figure 10 ijms-23-14022-f010:**
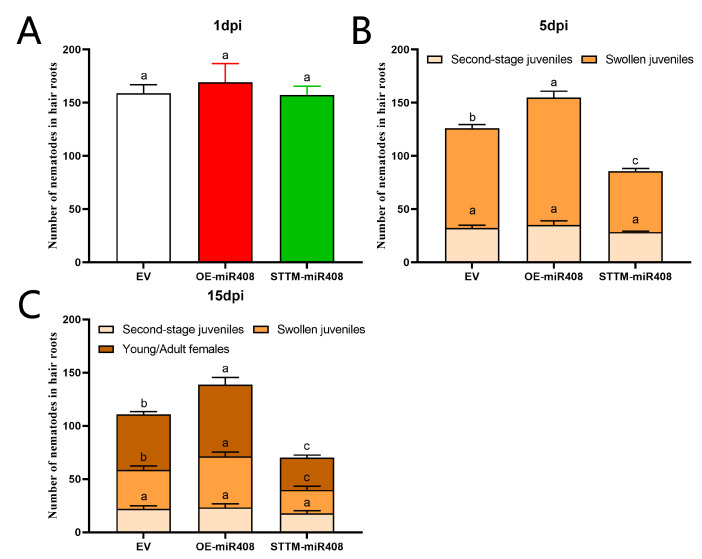
Infection and development of soybean cyst nematode in transgenic hairy roots. (**A**) Number of second-stage juveniles (J2s) in soybean hairy roots examined at 1 dpi. (**B**) Number of different stages of nematodes in soybean hairy roots at 5 dpi. (**C**) Number of different stages of nematodes in soybean hairy roots at 15 dpi. Multiple statistical comparisons between nematodes in EV and other transgenic hairy roots were made by one-way ANOVA. Different characters mean significant differences found at *p* < 0.05.

**Table 1 ijms-23-14022-t001:** *Gma-miR408* family members described in miRbase and sRNAanno.

miRNA Name	Location	Mature Sequence-5p	Mature Sequence-3p
*gma-miR408*a	Chr2:837419-837550	CAGGGGAACAGGCAGAGCAUG	AUGCACUGCCUCUUCCCUGGC
*gma-miR408*b	Chr3:44626692-44626830	CUGGGAACAGGCAGGGCACGA	AUGCACUGCCUCUUCCCUGGC
*gma-miR408*c	Chr10:36557001-36557134	CAGGGGAACAGGCAGAGCAUG	AUGCACUGCCUCUUCCCUGGC
*gma-miR408*d	Chr19:47163684-47163821	CUGGGAACAGGCAGAGCACGA	AUGCACUGCCUCUUCCCUGGC

**Table 2 ijms-23-14022-t002:** Precursor sequences of *gma-miR408* family members described in sRNAanno.

miRNA Name	Precursor Sequence
*gma-miR408*a	ACAGGACAAAGCAGGGGAACAGGCAGAGCAUGGAUGGAGCUAUCAA
	CACAAUAUUGUCAAGAAACUGAGAGUGAGAGGAGAAAUAUGUUGUG
	GUUCUGCUCAUGCACUGCCUCUUCCCUGGCUCUGUCUCCA
*gma-miR408*b	CAGGACAAAGGCUGGGAACAGGCAGGGCACGAAUGGAGCUAUCAAC
	AGAAAAUGGUAAAGUGAGAAUGAAAGGAGAGAGAGAGAGAGAGAUC
	UGUUGUGGCUACGCUCAUGCACUGCCUCUUCCCUGGCUCUGUCUCUU
*gma-miR408*c	ACAGGACAAAGCAGGGGAACAGGCAGAGCAUGGAUGGAGCUAUCAA
	CACAAUAUUGUCAAGAAAGUGAGAAAGUGAGAGGAGAAAUCUGUUG
	UGGUUCUGCUCAUGCACUGCCUCUUCCCUGGCUUUGUCUCUA
*gma-miR408*d	CAGGGCAAAGGCUGGGAACAGGCAGAGCACGAAUGGAGCUAUCAAC
	AGAAAAUGGAGAAGUGAGAAAUGAGAGGAGAGAAAGAGAGAGAUC
	UGUUAGGGCUACGCUCAUGCACUGCCUCUUCCCUGGCUCUGUCUCUU

**Table 3 ijms-23-14022-t003:** The raw reads and clean reads obtained in 9 small RNA libraries.

ID	Raw Reads	Clean Reads	Mapped Reads (%)	Uniquely M	Multiple M	Q30(%)
EV-1	46,804,494	39,360,542	36,609,992 (93.01)	35,970,124	639,868	90.89
EV-2	48,333,980	41,035,690	38,123,699 (92.90)	37,471,922	651,777	91.34
EV-3	48,613,286	40,137,396	37,364,825 (93.09)	36,697,461	667,364	91.25
OE-1	48,044,828	38,188,248	35,289,502 (92.41)	34,698,477	591,025	91.20
OE-2	56,847,288	45,955,306	42,646,102 (92.80)	41,945,756	700,346	91.84
OE-3	57,874,234	52,788,600	49,278,779 (93.35)	48,478,778	800,001	92.68
STTM-1	47,124,578	38,307,708	35,646,187 (93.05)	35,042,320	603,867	91.42
STTM-2	50,267,440	40,530,212	37,142,081 (91.64)	36,466,073	676,008	91.55
STTM-3	48,159,886	38,251,754	34,301,873 (89.67)	33,70,2718	599,155	91.13
Total	452,070,014	374,555,456	346,403,040	340,473,629	5,929,411	

**Table 4 ijms-23-14022-t004:** The number of differentially expressed genes in each comparison group.

Group	Total	Down Regulated	Up Regulate
EV vs OE	3653	1462	2191
EV vs STTM	933	335	598
OE vs STTM	3831	2205	1626

## Data Availability

The data presented in this study are available on request from the corresponding author.

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
