# Peer review of "Gma-miR408 Enhances Soybean Cyst Nematode Susceptibility by Suppressing Reactive Oxygen Species Accumulation"

_ijms, 2022, doi:10.3390/ijms232214022_

Round 1

Reviewer 1 Report

Thank you for this interesting research work. The manuscript is well structured and should be considered for publication in IJMS after some minor revisions. Please check my following comments:

L19: there should be a space between levels and [1].

L47: it should be [7-10].

L56: there is no need “,” after author name (Ma).

L79: “A” should be written in lowercase letter.

L108: it should be “…(qRT-PCR) among…

L110 and L111: it should be “non-infected plants”

L114: it should be “… were not different between…”

Figure 5 caption: the analysis method applied in Figure 5 is one-way ANOVA or t-test? Please confirm.

L154: “Different” should be written in lowercase letter.

Please unify p<0.05 throughout the text. Sometimes “p” is written in italic, sometimes is not in italic.

Discussion part is cursory and repeats the results. It should be improved or combined with the results.

Author Response

Response to Reviewer 1 Comments

Point 1: Please check my following comments:

L19: there should be a space between levels and [1].

L47: it should be [7-10].

L56: there is no need “,” after author name (Ma).

L79: “A” should be written in lowercase letter.

L108: it should be “…(qRT-PCR) among…

L110 and L111: it should be “non-infected plants”

L114: it should be “… were not different between…”

Figure 5 caption: the analysis method applied in Figure 5 is one-way ANOVA or t-test? Please confirm.

L154: “Different” should be written in lowercase letter.

Please unify p<0.05 throughout the text. Sometimes “p” is written in italic, sometimes is not in italic.

Response 1: I have read the full text and proofread it carefully to ensure consistency in content and format.

Point 2: Discussion part is cursory and repeats the results. It should be improved or combined with the results.

Response 2: Condensed the results of the Discussion section and added knowledge about the role of miRNA408 regulation in the processes of response to biotic and abiotic stresses.

Reviewer 2 Report

Dear authors.

The manuscript entitled "Gma-miR408 Enhances Soybean Cyst Nematode Susceptibility by Suppressing Reactive Oxygen Species Accumulation" is an interesting description of the research carried out. In my opinion, it seems to be correct both in terms of content and language. The conclusions drawn seem to be confirmed by the results of the research. Knowledge about the role of MicroRNA408 regulation in the processes of response to pathogens such as nematodes in plants of industrial importance should be expanded.

However, I have a few minor comments:

  Abbreviations should be explained immediately the first time you use them. This makes the manuscript easier to read by non-experts on the subject. For example: "J2s" line 190,

Fig 3, Fig 4, explain SCN and CK in the figure description.

Yours faithfully,

Author Response

Response to Reviewer 2 Comments

Point 1: Abbreviations should be explained immediately the first time you use them. This makes the manuscript easier to read by non-experts on the subject. For example: "J2s" line 190,

Fig 3, Fig 4, explain SCN and CK in the figure description.

Response 1: I have read the full text and proofread it carefully to makes the manuscript easier to read by non-experts on the subject.

Point 2: Knowledge about the role of MicroRNA408 regulation in the processes of response to pathogens such as nematodes in plants of industrial importance should be expanded.

Response 2: No further articles found on the role of miR408 in nematodes infection of important industrial plants. I added knowledge about the role of miRNA408 regulation in the processes of response to biotic and abiotic stresses.